# Machine Learning with Adversaries:
# Byzantine Tolerant Gradient Descent

**Peva Blanchard**
EPFL, Switzerland
peva.blanchard@epfl.ch

**El Mahdi El Mhamdi**\*
EPFL, Switzerland
elmahdi.elmhamdi@epfl.ch

**Rachid Guerraoui**
EPFL, Switzerland
rachid.guerraoui@epfl.ch

**Julien Stainer**
EPFL, Switzerland
julien.stainer@epfl.ch

## Abstract

We study the resilience to Byzantine failures of distributed implementations of Stochastic Gradient Descent (SGD). So far, distributed machine learning frameworks have largely ignored the possibility of failures, especially arbitrary (i.e., Byzantine) ones. Causes of failures include software bugs, network asynchrony, biases in local datasets, as well as attackers trying to compromise the entire system. Assuming a set of $n$ workers, up to $f$ being Byzantine, we ask how resilient can SGD be, without limiting the dimension, nor the size of the parameter space. We first show that no gradient aggregation rule based on a linear combination of the vectors proposed by the workers (i.e, current approaches) tolerates a single Byzantine failure. We then formulate a resilience property of the aggregation rule capturing the basic requirements to guarantee convergence despite $f$ Byzantine workers. We propose *Krum*, an aggregation rule that satisfies our resilience property, which we argue is the first provably Byzantine-resilient algorithm for distributed SGD. We also report on experimental evaluations of Krum.

## 1 Introduction

The increasing amount of data available [6], together with the growing complexity of machine learning models [27], has led to learning schemes that require a lot of computational resources. As a consequence, most industry-grade machine-learning implementations are now distributed [1]. For example, as of 2012, Google reportedly used 16.000 processors to train an image classifier [22]. More recently, attention has been given to federated learning and federated optimization settings [15, 16, 23] with a focus on communication efficiency. However, distributing a computation over several machines (worker processes) induces a higher risk of failures. These include crashes and computation errors, stalled processes, biases in the way the data samples are distributed among the processes, but also, in the worst case, attackers trying to compromise the entire system. The most robust system is one that tolerates *Byzantine* failures [17], i.e., completely arbitrary behaviors of some of the processes.

A classical approach to mask failures in distributed systems is to use a state machine replication protocol [26], which requires however state transitions to be applied by all worker processes. In the case of distributed machine learning, this constraint can be translated in two ways: either *(a)* the processes agree on a sample of data based on which they update their local parameter vectors, or *(b)* they agree on how the parameter vector should be updated. In case *(a)*, the sample of data has to be transmitted to each process, which then has to perform a heavyweight computation to update its local

parameter vector. This entails communication and computational costs that defeat the entire purpose of distributing the work. In case *(b)*, the processes have no way to check if the chosen update for the parameter vector has indeed been computed correctly on real data: a Byzantine process could have proposed the update and may easily prevent the convergence of the learning algorithm. Neither of these solutions is satisfactory in a realistic distributed machine learning setting.

In fact, most learning algorithms today rely on a core component, namely *stochastic gradient descent* (SGD) [4, 13], whether for training neural networks [13], regression [34], matrix factorization [12] or support vector machines [34]. In all those cases, a cost function – depending on the parameter vector – is minimized based on stochastic estimates of its gradient. Distributed implementations of SGD [33] typically take the following form: a single parameter server is in charge of updating the parameter vector, while worker processes perform the actual update estimation, based on the share of data they have access to. More specifically, the parameter server executes learning rounds, during each of which, the parameter vector is broadcast to the workers. In turn, each worker computes an estimate of the update to apply (an estimate of the *gradient*), and the parameter server aggregates their results to finally update the parameter vector. Today, this aggregation is typically implemented through averaging [25], or variants of it [33, 18, 31].

This paper addresses the fundamental question of how a distributed SGD can be devised to tolerate $f$ Byzantine processes among the $n$ workers.

**Contributions.** We first show in this paper that no linear combination (current approaches) of the updates proposed by the workers can tolerate a *single* Byzantine worker. Basically, a single Byzantine worker can force the parameter server to choose any arbitrary vector, even one that is too large in amplitude or too far in direction from the other vectors. Clearly, the Byzantine worker can prevent any classic averaging-based approach to converge. Choosing the appropriate aggregation of the vectors proposed by the workers turns out to be challenging. A non-linear, *squared-distance-based* aggregation rule, that selects, among the proposed vectors, the vector "closest to the barycenter" (for example by taking the vector that minimizes the sum of the squared distances to every other vector), might look appealing. Yet, such a squared-distance-based aggregation rule tolerates only a single Byzantine worker. Two Byzantine workers can collude, one helping the other to be selected, by moving the barycenter of all the vectors farther from the "correct area".

We formulate a Byzantine resilience property capturing sufficient conditions for the parameter server's aggregation rule to tolerate $f$ Byzantine workers. Essentially, to guarantee that the cost will decrease despite Byzantine workers, we require the vector output chosen by the parameter server *(a)* to point, on average, to the same direction as the gradient and *(b)* to have statistical moments (up to the fourth moment) bounded above by a homogeneous polynomial in the moments of a correct estimator of the gradient. One way to ensure such a resilience property is to consider a *majority-based* approach, looking at every subset of $n - f$ vectors, and considering the subset with the smallest diameter. While this approach is more robust to Byzantine workers that propose vectors far from the correct area, its exponential computational cost is prohibitive. Interestingly, combining the intuitions of the *majority-based* and *squared-distance* [2]*-based* methods, we can choose the vector that is somehow the closest to its $n - f$ neighbors. Namely, the one that minimizes the sum of squared distances to its $n - f$ closest vectors. This is the main idea behind our aggregation rule, we call *Krum*[3]. Assuming $2f + 2 < n$, we show that Krum satisfies the resilience property aforementioned and the corresponding machine learning scheme converges. An important advantage of Krum is its (local) time complexity ($O(n^2 \cdot d)$), linear in the dimension of the gradient, where $d$ is the dimension of the parameter vector. (In modern machine learning, the dimension $d$ of the parameter vector may take values in the hundreds of billions [30].) For simplicity of presentation, the version of Krum we first consider selects only one vector. We also discuss other variants.

We evaluate Krum experimentally, and compare it to classical averaging. We confirm the very fact that averaging does not stand Byzantine attacks, while Krum does. In particular, we report on attacks by omniscient adversaries – aware of a good estimate of the gradient – that send the opposite vector multiplied by a large factor, as well as attacks by adversaries that send random vectors drawn from a Gaussian distribution (the larger the variance of the distribution, the stronger the attack). We also

evaluate the extent to which Krum might slow down learning (compared to averaging) when there are no Byzantine failures. Interestingly, as we show experimentally, this slow down occurs only when the mini-batch size is close to 1. In fact, the slowdown can be drastically reduced by choosing a reasonable mini-batch size. We also evaluate *Multi-Krum*, a variant of Krum, which, intuitively, interpolates between Krum and averaging, thereby allowing to mix the resilience properties of Krum with the convergence speed of averaging. Multi-Krum outperforms other aggregation rules like the medoid, inspired by the geometric median.

**Paper Organization.** Section 2 recalls the classical model of distributed SGD. Section 3 proves that linear combinations (solutions used today) are not resilient even to a single Byzantine worker, then introduces our new concept of $(\alpha, f)$-Byzantine resilience. Section 4 introduces our Krum function, computes its computational cost and proves its $(\alpha, f)$-Byzantine resilience. Section 5 analyzes the convergence of a distributed SGD using Krum. Section 6 presents our experimental evaluation of Krum. We discuss related work and open problems in Section 7. Due to space limitations, some proofs and complementary experimental results are given as supplementary material.

## 2 Model

We consider the general distributed system model of [1], consisting of a parameter server[4], and $n$ workers, $f$ of them possibly Byzantine (behaving arbitrarily). Computation is divided into (infinitely many) synchronous rounds. During round $t$, the parameter server broadcasts its parameter vector $x_t \in \mathbb{R}^d$ to all the workers. Each correct worker $p$ computes an estimate $V_p^t = G(x_t, \xi_p^t)$ of the gradient $\nabla Q(x_t)$ of the cost function $Q$, where $\xi_p^t$ is a random variable representing, e.g., the sample (or a mini-batch of samples) drawn from the dataset. A Byzantine worker $b$ proposes a vector $V_b^t$ which can deviate arbitrarily from the vector it is supposed to send if it was correct, i.e., according to the algorithm assigned to it by the system developer (see Figure 1).

Since the communication is synchronous, if the parameter server does not receive a vector value $V_b^t$ from a given Byzantine worker $b$, then the parameter server acts as if it had received the default value $V_b^t = 0$ instead.

The parameter server computes a vector $F(V_1^t, \ldots, V_n^t)$ by applying a deterministic function $F$ (aggregation rule) to the vectors received. We refer to $F$ as the *aggregation rule* of the parameter server. The parameter server updates the parameter vector using the following SGD equation

$$x_{t+1} = x_t - \gamma_t \cdot F(V_1^t, \ldots, V_n^t).$$

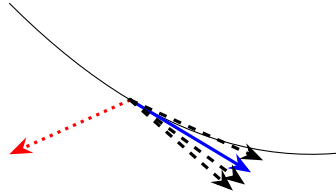

Figure 1: The gradient estimates computed by correct workers (black dashed arrows) are distributed around the actual gradient (solid arrow) of the cost function (thin black curve). A Byzantine worker can propose an arbitrary vector (red dotted arrow).

The correct (non-Byzantine) workers are assumed to compute unbiased estimates of the gradient $\nabla Q(x_t)$. More precisely, in every round $t$, the vectors $V_i^t$'s proposed by the correct workers are independent identically distributed random vectors, $V_i^t \sim G(x_t, \xi_i^t)$ with $\mathbb{E}_{\xi_i^t} G(x_t, \xi_i^t) = \nabla Q(x_t)$. This can be achieved by ensuring that each sample of data used for computing the gradient is drawn uniformly and independently, as classically assumed in the literature of machine learning [3].

The Byzantine workers have full knowledge of the system, including the aggregation rule $F$ as well as the vectors proposed by the workers. They can furthermore collaborate with each other [21].

## 3 Byzantine Resilience

In most SGD-based learning algorithms used today [4, 13, 12], the aggregation rule consists in computing the average [5] of the input vectors. Lemma 1 below states that no linear combination of the vectors can tolerate a single Byzantine worker. In particular, averaging is not Byzantine resilient.

**Lemma 1.** *Consider an aggregation rule $F_{lin}$ of the form $F_{lin}(V_1, \ldots, V_n) = \sum_{i=1}^{n} \lambda_i \cdot V_i$, where the $\lambda_i$'s are non-zero scalars. Let $U$ be any vector in $\mathbb{R}^d$. A single Byzantine worker can make $F$ always select $U$. In particular, a single Byzantine worker can prevent convergence.*

*Proof.* Immediate: if the Byzantine worker proposes $V_n = \frac{1}{\lambda_n} \cdot U - \sum_{i=1}^{n-1} \frac{\lambda_i}{\lambda_n} V_i$, then $F = U$.[6]  □

In the following, we define basic requirements on an appropriate Byzantine-resilient aggregation rule. Intuitively, the aggregation rule should output a vector $F$ that is not too far from the "real" gradient $g$, more precisely, the vector that points to the steepest direction of the cost function being optimized. This is expressed as a lower bound (condition *(i)*) on the scalar product of the (expected) vector $F$ and $g$. Figure 2 illustrates the situation geometrically. If $\mathbb{E}F$ belongs to the ball centered at $g$ with radius $r$, then the scalar product is bounded below by a term involving $\sin \alpha = r/\|g\|$.

Condition *(ii)* is more technical, and states that the moments of $F$ should be controlled by the moments of the (correct) gradient estimator $G$. The bounds on the moments of $G$ are classically used to control the effects of the discrete nature of the SGD dynamics [3]. Condition *(ii)* allows to transfer this control to the aggregation rule.

**Definition 1** (($\alpha, f$)-Byzantine Resilience)**.** *Let $0 \leq \alpha < \pi/2$ be any angular value, and any integer $0 \leq f \leq n$. Let $V_1, \ldots, V_n$ be any independent identically distributed random vectors in $\mathbb{R}^d$, $V_i \sim G$, with $\mathbb{E}G = g$. Let $B_1, \ldots, B_f$ be any random vectors in $\mathbb{R}^d$, possibly dependent on the $V_i$'s. aggregation rule $F$ is said to be ($\alpha, f$)-Byzantine resilient if, for any $1 \leq j_1 < \cdots < j_f \leq n$, vector*

$$F = F(V_1, \ldots, \underbrace{B_1}_{j_1}, \ldots, \underbrace{B_f}_{j_f}, \ldots, V_n)$$

*satisfies* (i) $\langle \mathbb{E}F, g \rangle \geq (1 - \sin \alpha) \cdot \|g\|^2 > 0$ *and* (ii) *for $r = 2, 3, 4$, $\mathbb{E} \|F\|^r$ is bounded above by a linear combination of terms $\mathbb{E} \|G\|^{r_1} \ldots \mathbb{E} \|G\|^{r_{n-1}}$ with $r_1 + \cdots + r_{n-1} = r$.*

## 4  The Krum Function

We now introduce *Krum*, our aggregation rule, which, we show, satisfies the ($\alpha, f$)-Byzantine resilience condition. The barycentric aggregation rule $F_{bary} = \frac{1}{n} \sum_{i=1}^{n} V_i$ can be defined as the vector in $\mathbb{R}^d$ that minimizes the sum of squared distances [7] to the $V_i$'s $\sum_{i=1}^{n} \|F_{bary} - V_i\|^2$. Lemma 1, however, states that this approach does not tolerate even a single Byzantine failure. One could try to select the vector $U$ *among* the $V_i$'s which minimizes the sum $\sum_i \|U - V_i\|^2$, i.e., which is "closest to all vectors".

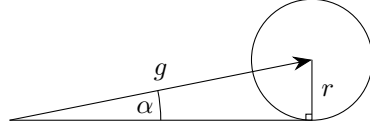

Figure 2: If $\|\mathbb{E}F - g\| \leq r$ then $\langle \mathbb{E}F, g \rangle$ is bounded below by $(1 - \sin \alpha)\|g\|^2$ where $\sin \alpha = r/\|g\|$.

However, because such a sum involves all the vectors, even those which are very far, this approach does not tolerate Byzantine workers: by proposing large enough vectors, a Byzantine worker can force the total barycenter to get closer to the vector proposed by another Byzantine worker.

Our approach to circumvent this issue is to preclude the vectors that are too far away. More precisely, we define our *Krum* aggregation rule $\text{KR}(V_1, \ldots, V_n)$ as follows. For any $i \neq j$, we denote by $i \to j$ the fact that $V_j$ belongs to the $n - f - 2$ closest vectors to $V_i$. Then, we define for each worker $i$, the *score* $s(i) = \sum_{i \to j} \|V_i - V_j\|^2$ where the sum runs over the $n - f - 2$ closest vectors to $V_i$. Finally, $\text{KR}(V_1, \ldots, V_n) = V_{i_*}$ where $i_*$ refers to the worker minimizing the score, $s(i_*) \leq s(i)$ for all $i$.[8]

**Lemma 2.** *The expected time complexity of the Krum Function $\text{KR}(V_1, \ldots, V_n)$, where $V_1, \ldots, V_n$ are $d$-dimensional vectors, is $O(n^2 \cdot d)$*

*Proof.* For each $V_i$, the parameter server computes the $n$ squared distances $\|V_i - V_j\|^2$ (time $O(n \cdot d)$). Then the parameter server selects the first $n - f - 1$ of these distances (expected time $O(n)$ with Quickselect) and sums their values (time $O(n \cdot d)$). Thus, computing the score of all the $V_i$'s takes $O(n^2 \cdot d)$. An additional term $O(n)$ is required to find the minimum score, but is negligible relatively to $O(n^2 \cdot d)$. $\qquad\square$

Proposition 1 below states that, if $2f + 2 < n$ and the gradient estimator is accurate enough, (its standard deviation is relatively small compared to the norm of the gradient), then the Krum function is $(\alpha, f)$-Byzantine-resilient, where angle $\alpha$ depends on the ratio of the deviation over the gradient.

**Proposition 1.** *Let $V_1, \ldots, V_n$ be any independent and identically distributed random d-dimensional vectors s.t $V_i \sim G$, with $\mathbb{E}G = g$ and $\mathbb{E}\|G - g\|^2 = d\sigma^2$. Let $B_1, \ldots, B_f$ be any $f$ random vectors, possibly dependent on the $V_i$'s. If $2f + 2 < n$ and $\eta(n, f)\sqrt{d} \cdot \sigma < \|g\|$, where*

$$\eta(n, f) \underset{def}{=} \sqrt{2\left(n - f + \frac{f \cdot (n - f - 2) + f^2 \cdot (n - f - 1)}{n - 2f - 2}\right)} = \begin{cases} O(n) & \text{if } f = O(n) \\ O(\sqrt{n}) & \text{if } f = O(1) \end{cases},$$

*then the Krum function* KR *is $(\alpha, f)$-Byzantine resilient where $0 \leq \alpha < \pi/2$ is defined by*

$$\sin \alpha = \frac{\eta(n, f) \cdot \sqrt{d} \cdot \sigma}{\|g\|}.$$

The condition on the norm of the gradient, $\eta(n, f) \cdot \sqrt{d} \cdot \sigma < \|g\|$, can be satisfied, to a certain extent, by having the (correct) workers compute their gradient estimates on mini-batches [3]. Indeed, averaging the gradient estimates over a mini-batch divides the deviation $\sigma$ by the squared root of the size of the mini-batch. For the sake of concision, we only give here the sketch of the proof. (We give the detailed proof in the supplementary material.)

*Proof.* (Sketch) Without loss of generality, we assume that the Byzantine vectors $B_1, \ldots, B_f$ occupy the last $f$ positions in the list of arguments of KR, i.e., KR $=$ KR$(V_1, \ldots, V_{n-f}, B_1, \ldots, B_f)$. Let $i_*$ be the index of the vector chosen by the Krum function. We focus on the condition *(i)* of $(\alpha, f)$-Byzantine resilience (Definition 1).

Consider first the case where $V_{i_*} = V_i \in \{V_1, \ldots, V_{n-f}\}$ is a vector proposed by a correct process. The first step is to compare the vector $V_i$ with the average of the correct vectors $V_j$ such that $i \to j$. Let $\delta_c(i)$ be the number of such $V_j$'s.

$$\mathbb{E}\left\|V_i - \frac{1}{\delta_c(i)}\sum_{i \to \text{ correct } j} V_j\right\|^2 \leq \frac{1}{\delta_c(i)}\sum_{i \to \text{ correct } j} \mathbb{E}\|V_i - V_j\|^2 \leq 2d\sigma^2. \tag{1}$$

The last inequality holds because the right-hand side of the first inequality involves only vectors proposed by correct processes, which are mutually independent and follow the distribution of $G$.

Now, consider the case where $V_{i_*} = B_k \in \{B_1, \ldots, B_f\}$ is a vector proposed by a Byzantine process. The fact that $k$ minimizes the score implies that for all indices $i$ of vectors proposed by correct processes

$$\sum_{k \to \text{ correct } j} \|B_k - V_j\|^2 + \sum_{k \to \text{ byz } l} \|B_k - B_l\|^2 \leq \sum_{i \to \text{ correct } j} \|V_i - V_j\|^2 + \sum_{i \to \text{ byz } l} \|V_i - B_l\|^2.$$

Then, for all indices $i$ of vectors proposed by correct processes

$$\left\|B_k - \frac{1}{\delta_c(k)}\sum_{k \to \text{ correct } j} V_j\right\|^2 \leq \frac{1}{\delta_c(k)}\sum_{i \to \text{ correct } j} \|V_i - V_j\|^2 + \frac{1}{\delta_c(k)}\underbrace{\sum_{i \to \text{ byz } l} \|V_i - B_l\|^2}_{D^2(i)}.$$

The term $D^2(i)$ is the only term involving vectors proposed by Byzantine processes. However, the correct process $i$ has $n - f - 2$ neighbors and $f + 1$ non-neighbors. Therefore, there exists a correct

process $\zeta(i)$ which is farther from $i$ than every neighbor $j$ of $i$ (including the Byzantine neighbors). In particular, for all $l$ such that $i \to l$, $\|V_i - B_l\|^2 \leq \|V_i - V_{\zeta(i)}\|^2$. Thus

$$\left\| B_k - \frac{1}{\delta_c(k)} \sum_{k \to \text{correct } j} V_j \right\|^2 \leq \frac{1}{\delta_c(k)} \sum_{i \to \text{correct } j} \|V_i - V_j\|^2 + \frac{n - f - 2 - \delta_c(i)}{\delta_c(k)} \|V_i - V_{\zeta(i)}\|^2. \tag{2}$$

Combining equations 1, 2, and a union bound yields $\|\mathbb{E}\text{KR} - g\|^2 \leq \eta\sqrt{d}\|g\|$, which, in turn, implies $\langle \mathbb{E}\text{KR}, g \rangle \geq (1 - \sin\alpha)\|g\|^2$. Condition *(ii)* is proven by bounding the moments of KR with moments of the vectors proposed by the correct processes only, using the same technique as above. The full proof is provided in the supplementary material. $\qquad\square$

## 5 Convergence Analysis

In this section, we analyze the convergence of the SGD using our Krum function defined in Section 4. The SGD equation is expressed as follows

$$x_{t+1} = x_t - \gamma_t \cdot \text{KR}(V_1^t, \dots, V_n^t)$$

where at least $n - f$ vectors among the $V_i^t$'s are correct, while the other ones may be Byzantine. For a correct index $i$, $V_i^t = G(x_t, \xi_i^t)$ where $G$ is the gradient estimator. We define the *local standard deviation* $\sigma(x)$ by

$$d \cdot \sigma^2(x) = \mathbb{E}\|G(x, \xi) - \nabla Q(x)\|^2.$$

The following proposition considers an (*a priori*) non-convex cost function. In the context of non-convex optimization, even in the centralized case, it is generally hopeless to aim at proving that the parameter vector $x_t$ tends to a local minimum. Many criteria may be used instead. We follow [3], and we prove that the parameter vector $x_t$ almost surely reaches a "flat" region (where the norm of the gradient is small), in a sense explained below.

**Proposition 2.** *We assume that* (i) *the cost function $Q$ is three times differentiable with continuous derivatives, and is non-negative, $Q(x) \geq 0$;* (ii) *the learning rates satisfy $\sum_t \gamma_t = \infty$ and $\sum_t \gamma_t^2 < \infty$;* (iii) *the gradient estimator satisfies $\mathbb{E}G(x, \xi) = \nabla Q(x)$ and $\forall r \in \{2, \dots, 4\}$, $\mathbb{E}\|G(x, \xi)\|^r \leq A_r + B_r\|x\|^r$ for some constants $A_r, B_r$;* (iv) *there exists a constant $0 \leq \alpha < \pi/2$ such that for all $x$*

$$\eta(n, f) \cdot \sqrt{d} \cdot \sigma(x) \leq \|\nabla Q(x)\| \cdot \sin\alpha;$$

(v) *finally, beyond a certain horizon, $\|x\|^2 \geq D$, there exist $\epsilon > 0$ and $0 \leq \beta < \pi/2 - \alpha$ such that $\|\nabla Q(x)\| \geq \epsilon > 0$ and $\frac{\langle x, \nabla Q(x) \rangle}{\|x\| \cdot \|\nabla Q(x)\|} \geq \cos\beta$. Then the sequence of gradients $\nabla Q(x_t)$ converges almost surely to zero.*

Conditions *(i)* to *(iv)* are the same conditions as in the non-convex convergence analysis in [3]. Condition *(v)* is a slightly stronger condition than the corresponding one in [3], and states that, beyond a certain horizon, the cost function $Q$ is "convex enough", in the sense that the direction of the gradient is sufficiently close to the direction of the parameter vector $x$. Condition *(iv)*, however, states that the gradient estimator used by the correct workers has to be accurate enough, i.e., the local standard deviation should be small relatively to the norm of the gradient.

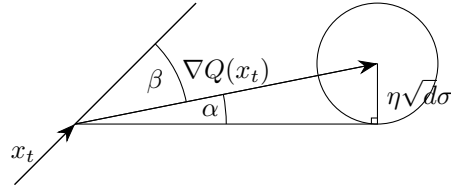

Figure 3: Condition on the angles between $x_t$, $\nabla Q(x_t)$ and $\mathbb{E}\text{KR}_t$, in the region $\|x_t\|^2 > D$.

Of course, the norm of the gradient tends to zero near, e.g., extremal and saddle points. Actually, the ratio $\eta(n, f) \cdot \sqrt{d} \cdot \sigma / \|\nabla Q\|$ controls the maximum angle between the gradient $\nabla Q$ and the vector chosen by the Krum function. In the regions where $\|\nabla Q\| < \eta(n, f) \cdot \sqrt{d} \cdot \sigma$, the Byzantine workers may take advantage of the noise (measured by $\sigma$) in the gradient estimator $G$ to bias the choice of the parameter server. Therefore, Proposition 2 is to be interpreted as follows: in the presence of Byzantine workers, the parameter vector $x_t$ almost surely reaches a basin around points where the gradient is small ($\|\nabla Q\| \leq \eta(n, f) \cdot \sqrt{d} \cdot \sigma$), i.e., points where the cost landscape is "almost flat".

Note that the convergence analysis is based only on the fact that function KR is $(\alpha, f)$-Byzantine resilient. The complete proof of Proposition 2 is deferred to the supplementary material.

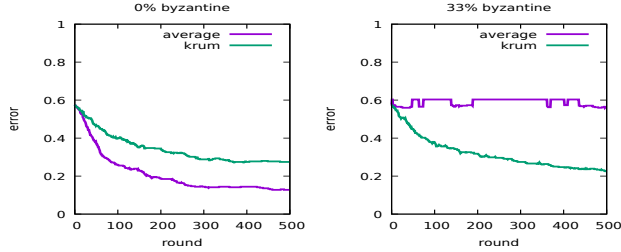

Figure 4: Cross-validation error evolution with rounds, respectively in the absence and in the presence of 33% Byzantine workers. The mini-batch size is 3. With 0% Gaussian Byzantine workers, averaging converges faster than Krum. With 33% Gaussian Byzantine workers, averaging does not converge, whereas Krum behaves as if there were 0% Byzantine workers.

## 6 Experimental Evaluation

We report here on the evaluation of the convergence and resilience properties of Krum, as well as an optimized variant of it. (We also discuss other variants of Krum in the supplementary material.)

**(Resilience to Byzantine processes).** We consider the task of spam filtering (dataset *spambase* [19]). The learning model is a multi-layer perceptron (MLP) with two hidden layers. There are $n = 20$ worker processes. Byzantine processes propose vectors drawn from a Gaussian distribution with mean zero, and isotropic covariance matrix with standard deviation 200. We refer to this behavior as *Gaussian Byzantine*. Each (correct) worker estimates the gradient on a mini-batch of size 3. We measure the error using cross-validation. Figure 4 shows how the error ($y$-axis) evolves with the number of rounds ($x$-axis).

In the first plot (left), there are no Byzantine workers. Unsurprisingly, averaging converges faster than Krum. In the second plot (right), 33% of the workers are Gaussian Byzantine. In this case, averaging does not converge at all, whereas Krum behaves as if there were no Byzantine workers. This experiment confirms that averaging does not tolerate (the rather mild) Gaussian Byzantine behavior, whereas Krum does.

**(The Cost of Resilience).** As seen above, Krum slows down learning when there are no Byzantine workers. The following experiment shows that this overhead can be significantly reduced by slightly increasing the mini-batch size. To highlight the effect of the presence of Byzantine workers, the Byzantine behavior has been set as follows: each Byzantine worker computes an estimate of the gradient over the *whole* dataset (yielding a very accurate estimate of the gradient), and proposes the opposite vector, scaled to a large length. We refer to this behavior as *omniscient*.

Figure 5 displays how the error value at the 500-th round ($y$-axis) evolves when the mini-batch size varies ($x$-axis). In this experiment, we consider the tasks of spam filtering (dataset *spambase*) and image classification (dataset *MNIST*). The MLP model is used in both cases. Each curve is obtained with either 0 or 45% of omniscient Byzantine workers.

In all cases, averaging still does not tolerate Byzantine workers, but yields the lowest error when there are no Byzantine workers. However, once the size of the mini-batch reaches the value 20, Krum with 45% omniscient Byzantine workers is as accurate as averaging with 0% Byzantine workers. We observe a similar pattern for a ConvNet as provided in the supplementary material.

**(Multi-Krum).** For the sake of presentation simplicity, we considered a version of Krum which selects only one vector among the vector proposed by the workers. We also propose a variant of Krum, we call *Multi-Krum*. Multi-Krum computes, for each vector proposed, the score as in the Krum function. Then, Multi-Krum selects the $m \in \{1, \dots, n\}$ vectors $V_1^*, \dots, V_m^*$ which score the best, and outputs their average $\frac{1}{m} \sum_i V_i^*$. Note that, the cases $m = 1$ and $m = n$ correspond to Krum and averaging respectively.

Figure 6 shows how the error ($y$-axis) evolves with the number of rounds ($x$-axis). In the figure, we consider the task of spam filtering (dataset *spambase*), and the MLP model (the same comparison

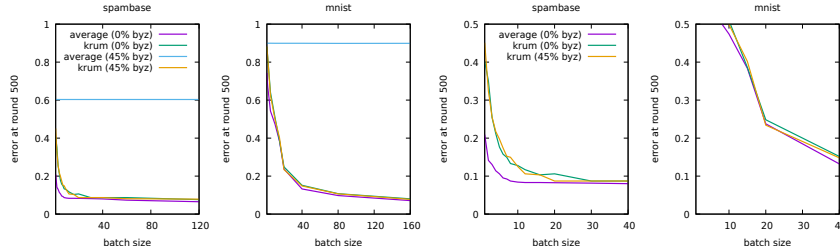

Figure 5: Cross-validation error at round 500 when increasing the mini-batch size. The two figures on the rights are zoomed versions of the two on the left). With a reasonably large mini-batch size (arround 10 for MNIST and 30 for Spambase), Krum with $45\%$ omniscient Byzantine workers is as accurate as averaging with $0\%$ Byzantine workers.

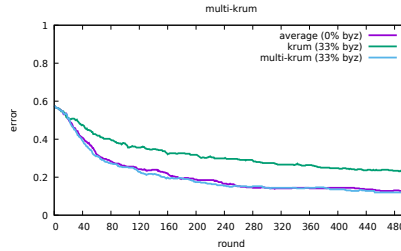

Figure 6: Cross-validation error evolution with rounds. The mini-batch size is 3. Multi-Krum with 33% Gaussian Byzantine workers converges as fast as averaging with 0% Byzantine workers.

is done for the task of image classification with a ConvNet and is provided in the supplementary material). The Multi-Krum parameter $m$ is set to $m = n - f$. Figure 6 shows that Multi-Krum with 33% Byzantine workers is as efficient as averaging with 0% Byzantine workers.

From the practitionner's perspective, the parameter $m$ may be used to set a specific trade-off between convergence speed and resilience to Byzantine workers.

## 7  Concluding Remarks

**(The Distributed Computing Perspective).**    Although seemingly related, results in $d$-dimensional approximate agreement [24, 14] cannot be applied to our Byzantine-resilient machine context for the following reasons: *(a)* [24, 14] assume that the set of vectors that can be proposed to an instance of the agreement is bounded so that at least $f + 1$ correct workers propose the same vector, which would require a lot of redundant work in our setting; and more importantly, *(b)* [24] requires a local computation by each worker that is in $O(n^d)$. While this cost seems reasonable for small dimensions, such as, e.g., mobile robots meeting in a $2D$ or $3D$ space, it becomes a real issue in the context of machine learning, where $d$ may be as high as 160 billion [30] (making $d$ a crucial parameter when considering complexities, either for local computations, or for communication rounds). The expected time complexity of the Krum function is $O(n^2 \cdot d)$. A closer approach to ours has been recently proposed in [28, 29]. In [28], the study only deals with parameter vectors of dimension one, which is too restrictive for today's multi-dimensional machine learning. In [29], the authors tackle a multi-dimensional situation, using an iterated approximate Byzantine agreement that reaches consensus asymptotically. This is however only achieved on a finite set of possible environmental states and cannot be used in the continuous context of stochastic gradient descent.

**(The Statistics and Machine Learning View).**    Our work looks at the resilience of the aggregation rule using ideas that are close to those of [11], and somehow classical in theoretical statistics on the robustness of the geometric median and the notion of breakdown [7]. The closest concept to a breakdown in our work is the maximum fraction of Byzantine workers that can be tolerated, i.e. $\frac{n-2}{2n}$, which reaches the optimal theoretical value $(1/2)$ asymptotically on $n$. It is known that the geometric

median does achieve the optimal breakdown. However, no closed form nor an exact algorithm to compute the geometric median is known (only approximations are available [5] and their Byzantine resilience is an open problem.). An easily computable variant of the median is the *Medoid*, which is the proposed vector minimizing the sum of distances to all other proposed vectors. The Medoid can be computed with a similar algorithm to Krum. We show however in the supplementary material that the implementation of the Medoid is outperformed by multi-Krum.

**(Robustness *Within* the Model).** It is important to keep in mind that this work deals with robustness from a coarse-grained perspective: the unit of failure is a worker, receiving its copy of the model and estimating gradients, based on either local data or delegated data from a server. The nature of the model itself is not important, the distributed system can be training models spanning a large range from simple regression to deep neural networks. As long as this training is using gradient-based learning, our algorithm to aggregate gradients, *Krum*, provably ensures convergence when a simple majority of nodes are not compromised by an attacker.

A natural question to consider is the fine-grained view: is the model itself robust to internal perturbations? In the case of neural networks, this question can somehow be tied to neuroscience considerations: could some neurons and/or synapses misbehave individually without harming the global outcome? We formulated this question in another work and proved a tight upper bound on the resulting global error when a set of nodes is removed or is misbehaving [8]. One of the many practical consequences [9] of such fine-grained view is the understanding of memory cost reduction trade-offs in deep learning. Such memory cost reduction can be viewed as the introduction of precision errors at the level of each neuron and/or synapse [8].

Other approaches to robustness within the model tackled adversarial situations in machine learning with a focus on adversarial examples (during inference) [10, 32, 11] instead of adversarial gradients (during training) as we did for *Krum*. Robustness to adversarial input can be viewed through the fine-grained lens we introduced in [8], for instance, one can see perturbations of pixels in the inputs as perturbations of neurons in layer zero. It is important to note the orthogonality and complementarity between the fine-grained (model/input units) and the coarse-grained (gradient aggregation) approaches. Being robust, as a model, either to adversarial examples or to internal perturbations, does not necessarily imply robustness to adversarial gradients during training. Similarly, being distributively trained with a robust aggregation scheme such as *Krum* does not necessarily imply robustness to internal errors of the model or adversarial input perturbations that would occur later during inference. For instance, the theory we develop in the present work is agnostic to the model being trained or the technology of the hardware hosting it, as long as there are gradients to be aggregated.

**Acknowledgment.** The authors would like to thank Georgios Damaskinos and Rhicheek Patra from the Distributed Computing group at EPFL for kindly providing their distributed machine learning framework, on top of which we could test our algorithm, *Krum*, and its variants described in this work. Further implementation details and additional experiments will be posted in the lab's Github repository [20]. The authors would also like to thank Saad Benjelloun, Lê Nguyen Hoang and Sébastien Rouault for fruitful comments. This work has been supported in part by the European ERC (Grant 339539 - AOC) and by the Swiss National Science Foundation (Grant 200021_ 169588 TARBDA). A preliminary version of this work appeared as a brief announcement during the $36^{st}$ ACM Symposium on Principles of Distributed Computing [2].

## Footnotes

\*contact author

[2]In all this paper, distances are computed with the Euclidean norm.

[3]Krum, in Greek Κρούμος, was a Bulgarian Khan of the end of the eighth century, who undertook offensive attacks against the Byzantine empire. Bulgaria doubled in size during his reign.

[4]The parameter server is assumed to be reliable. Classical techniques of state-machine replication can be used to ensure this.

[5]Or a closely related rule.

[6]Note that the parameter server could cancel the effects of the Byzantine behavior by setting, for example, $\lambda_n$ to 0. This however requires means to detect which worker is Byzantine.

[7]Removing the square of the distances leads to the geometric median, we discuss this when optimizing Krum.

[8]If two or more workers have the minimal score, we choose the one with the smallest identifier.

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
