[Supplementary Material · sgd_byz_supplementary_material(3).pdf]

# Machine Learning with Adversaries:
# Byzantine Tolerant Gradient Descent

## (Supplementary Material)

**Peva Blanchard**
EPFL, Switzerland
peva.blanchard@epfl.ch

**El Mahdi El Mhamdi**[*]
EPFL, Switzerland
elmahdi.elmhamdi@epfl.ch

**Rachid Guerraoui**
EPFL, Switzerland
rachid.guerraoui@epfl.ch

**Julien Stainer**
EPFL, Switzerland
julien.stainer@epfl.ch

## Abstract

In this document, we provide the complete proofs of the lemmas and propositions from the main paper *Machine Learning with Adversaries: Byzantine Tolerant Gradient Descent*. We also give the details of our variants of Krum aggregation rule and the geometric median, and compare them. Finally, we provide additional experimental results.

## 1  Byzantine Resilience

**Lemma 1.** *Consider a choice function $F_{lin}$ of the form $F_{lin}(V_1, \ldots, V_n) = \sum_{i=1}^{n} \lambda_i \cdot V_i$, where the $\lambda_i$'s are non-zero scalars. Let $U$ be any vector in $\mathbb{R}^d$. A single Byzantine worker can make $F$ always select $U$. In particular, a single Byzantine worker can prevent convergence.*

*Proof.* If the Byzantine worker proposes vector $V_n = \frac{1}{\lambda_n} \cdot U - \sum_{i=1}^{n-1} \frac{\lambda_i}{\lambda_n} V_i$, then $F = U$. Note that the parameter server could cancel the effects of the Byzantine behavior by setting, for example, $\lambda_n$ to 0, but this requires means to detect which worker is Byzantine. $\square$

## 2  The Krum Function

**Lemma 2.** *The expected time complexity of the Krum Function $\mathrm{KR}(V_1, \ldots, V_n)$, where $V_1, \ldots, V_n$ are $d$-dimensional vectors, is $O(n^2 \cdot d)$*

*Proof.* For each $V_i$, the parameter server computes the $n$ squared distances $\|V_i - V_j\|^2$ (time $O(n \cdot d)$). Then the parameter server selects the first $n - f - 1$ of these distances (expected time $O(n)$ with Quickselect) and sums their values (time $O(n \cdot d)$). Thus, computing the score of all the $V_i$'s takes $O(n^2 \cdot d)$. An additional term $O(n)$ is required to find the minimum score, but is negligible relatively to $O(n^2 \cdot d)$. $\square$

**Proposition 1.** *Let $V_1, \ldots, V_n$ be any independent and identically distributed random $d$-dimensional vectors s.t $V_i \sim G$, with $\mathbb{E}G = g$ and $\mathbb{E}\|G - g\|^2 = d\sigma^2$. Let $B_1, \ldots, B_f$ be any $f$ random vectors,*

---

[*]contact author

*possibly dependent on the $V_i$'s. If $2f + 2 < n$ and $\eta(n, f)\sqrt{d} \cdot \sigma < \|g\|$, where*

$$\eta(n, f) \underset{def}{=} \sqrt{2\left(n - f + \frac{f \cdot (n - f - 2) + f^2 \cdot (n - f - 1)}{n - 2f - 2}\right)} = \begin{cases} O(n) & \text{if } f = O(n) \\ O(\sqrt{n}) & \text{if } f = O(1) \end{cases},$$

*then the Krum function* $\mathrm{KR}$ *is* $(\alpha, f)$-*Byzantine resilient where* $0 \le \alpha < \pi/2$ *is defined by*

$$\sin\alpha = \frac{\eta(n, f) \cdot \sqrt{d} \cdot \sigma}{\|g\|}.$$

*Proof.* Without loss of generality, we assume that the Byzantine vectors $B_1, \ldots, B_f$ occupy the last $f$ positions in the list of arguments of $\mathrm{KR}$, i.e., $\mathrm{KR} = \mathrm{KR}(V_1, \ldots, V_{n-f}, B_1, \ldots, B_f)$. An index is *correct* if it refers to a vector among $V_1, \ldots, V_{n-f}$. An index is *Byzantine* if it refers to a vector among $B_1, \ldots, B_f$. For each index (correct or Byzantine) $i$, we denote by $\delta_c(i)$ (resp. $\delta_b(i)$) the number of correct (resp. Byzantine) indices $j$ such that $i \to j$. We have

$$\delta_c(i) + \delta_b(i) = n - f - 2$$
$$n - 2f - 2 \le \delta_c(i) \le n - f - 2$$
$$\delta_b(i) \le f.$$

We focus first on the condition *(i)* of $(\alpha, f)$-Byzantine resilience. We determine an upper bound on the squared distance $\|\mathbb{E}\mathrm{KR} - g\|^2$. Note that, for any correct $j$, $\mathbb{E}V_j = g$. We denote by $i_*$ the index of the vector chosen by the Krum function.

$$\|\mathbb{E}\mathrm{KR} - g\|^2 \le \left\| \mathbb{E}\left(\mathrm{KR} - \frac{1}{\delta_c(i_*)} \sum_{i_* \to \text{correct } j} V_j\right) \right\|^2$$

$$\le \mathbb{E}\left\| \mathrm{KR} - \frac{1}{\delta_c(i_*)} \sum_{i_* \to \text{correct } j} V_j \right\|^2 \quad \text{(Jensen inequality)}$$

$$\le \sum_{\text{correct } i} \mathbb{E}\left\| V_i - \frac{1}{\delta_c(i)} \sum_{i \to \text{correct } j} V_j \right\|^2 \mathbb{I}(i_* = i)$$

$$+ \sum_{\text{byz } k} \mathbb{E}\left\| B_k - \frac{1}{\delta_c(k)} \sum_{k \to \text{correct } j} V_j \right\|^2 \mathbb{I}(i_* = k)$$

where $\mathbb{I}$ denotes the indicator function[2]. We examine the case $i_* = i$ for some correct index $i$.

$$\left\| V_i - \frac{1}{\delta_c(i)} \sum_{i \to \text{correct } j} V_j \right\|^2 = \left\| \frac{1}{\delta_c(i)} \sum_{i \to \text{correct } j} V_i - V_j \right\|^2$$

$$\le \frac{1}{\delta_c(i)} \sum_{i \to \text{correct } j} \|V_i - V_j\|^2 \quad \text{(Jensen inequality)}$$

$$\mathbb{E}\left\| V_i - \frac{1}{\delta_c(i)} \sum_{i \to \text{correct } j} V_j \right\|^2 \le \frac{1}{\delta_c(i)} \sum_{i \to \text{correct } j} \mathbb{E}\|V_i - V_j\|^2$$

$$\le 2d\sigma^2.$$

We now examine the case $i_* = k$ for some Byzantine index $k$. The fact that $k$ minimizes the score implies that for all correct indices $i$

$$\sum_{k \to \text{correct } j} \|B_k - V_j\|^2 + \sum_{k \to \text{byz } l} \|B_k - B_l\|^2 \le \sum_{i \to \text{correct } j} \|V_i - V_j\|^2 + \sum_{i \to \text{byz } l} \|V_i - B_l\|^2.$$

Then, for all correct indices $i$

$$\left\| B_k - \frac{1}{\delta_c(k)} \sum_{k \to \text{ correct } j} V_j \right\|^2 \leq \frac{1}{\delta_c(k)} \sum_{k \to \text{ correct } j} \| B_k - V_j \|^2$$

$$\leq \frac{1}{\delta_c(k)} \sum_{i \to \text{ correct } j} \| V_i - V_j \|^2 + \frac{1}{\delta_c(k)} \underbrace{\sum_{i \to \text{ byz } l} \| V_i - B_l \|^2}_{D^2(i)}.$$

We focus on the term $D^2(i)$. Each correct process $i$ has $n - f - 2$ neighbors, and $f + 1$ non-neighbors. Thus there exists a correct worker $\zeta(i)$ which is farther from $i$ than any of the neighbors of $i$. In particular, for each Byzantine index $l$ such that $i \to l$, $\| V_i - B_l \|^2 \leq \| V_i - V_{\zeta(i)} \|^2$. Whence

$$\left\| B_k - \frac{1}{\delta_c(k)} \sum_{k \to \text{ correct } j} V_j \right\|^2 \leq \frac{1}{\delta_c(k)} \sum_{i \to \text{ correct } j} \| V_i - V_j \|^2 + \frac{\delta_b(i)}{\delta_c(k)} \| V_i - V_{\zeta(i)} \|^2$$

$$\mathbb{E} \left\| B_k - \frac{1}{\delta_c(k)} \sum_{k \to \text{ correct } j} V_j \right\|^2 \leq \frac{\delta_c(i)}{\delta_c(k)} \cdot 2 d \sigma^2 + \frac{\delta_b(i)}{\delta_c(k)} \sum_{\text{correct } j \neq i} \mathbb{E} \| V_i - V_j \|^2 \, \mathbb{I}(\zeta(i) = j)$$

$$\leq \left( \frac{\delta_c(i)}{\delta_c(k)} \cdot + \frac{\delta_b(i)}{\delta_c(k)} (n - f - 1) \right) 2 d \sigma^2$$

$$\leq \left( \frac{n - f - 2}{n - 2f - 2} + \frac{f}{n - 2f - 2} \cdot (n - f - 1) \right) 2 d \sigma^2.$$

Putting everything back together, we obtain

$$\| \mathbb{E} \text{KR} - g \|^2 \leq (n - f) 2 d \sigma^2 + f \cdot \left( \frac{n - f - 2}{n - 2f - 2} + \frac{f}{n - 2f - 2} \cdot (n - f - 1) \right) 2 d \sigma^2$$

$$\leq \underbrace{2 \left( n - f + \frac{f \cdot (n - f - 2) + f^2 \cdot (n - f - 1)}{n - 2f - 2} \right) d \sigma^2}_{\eta^2(n, f)}.$$

By assumption, $\eta(n, f) \sqrt{d} \sigma < \| g \|$, i.e., $\mathbb{E} \text{KR}$ belongs to a ball centered at $g$ with radius $\eta(n, f) \cdot \sqrt{d} \cdot \sigma$. This implies

$$\langle \mathbb{E} \text{KR}, g \rangle \geq \left( \| g \| - \eta(n, f) \cdot \sqrt{d} \cdot \sigma \right) \cdot \| g \| = (1 - \sin \alpha) \cdot \| g \|^2.$$

To sum up, condition (i) of the $(\alpha, f)$-Byzantine resilience property holds. We now focus on condition (ii).

$$\mathbb{E} \| \text{KR} \|^r = \sum_{\text{correct } i} \mathbb{E} \| V_i \|^r \, \mathbb{I}(i_* = i) + \sum_{\text{byz } k} \mathbb{E} \| B_k \|^r \, \mathbb{I}(i_* = k)$$

$$\leq (n - f) \mathbb{E} \| G \|^r + \sum_{\text{byz } k} \mathbb{E} \| B_k \|^r \, \mathbb{I}(i_* = k).$$

Denoting by $C$ a generic constant, when $i_* = k$, we have for all correct indices $i$

$$\left\| B_k - \frac{1}{\delta_c(k)} \sum_{k \to \text{correct } j} V_j \right\| \leq \sqrt{\frac{1}{\delta_c(k)} \sum_{i \to \text{ correct } j} \| V_i - V_j \|^2 + \frac{\delta_b(i)}{\delta_c(k)} \| V_i - V_{\zeta(i)} \|^2}$$

$$\leq C \cdot \left( \sqrt{\frac{1}{\delta_c(k)}} \cdot \sum_{i \to \text{correct } j} \| V_i - V_j \| + \sqrt{\frac{\delta_b(i)}{\delta_c(k)}} \cdot \| V_i - V_{\zeta(i)} \| \right)$$

$$\leq C \cdot \sum_{\text{correct } j} \| V_j \| \quad \text{(triangular inequality)}.$$

The second inequality comes from the equivalence of norms in finite dimension. Now

$$\|B_k\| \leq \left\| B_k - \frac{1}{\delta_c(k)} \sum_{k \to \text{correct } j} V_j \right\| + \left\| \frac{1}{\delta_c(k)} \sum_{k \to \text{correct } j} V_j \right\|$$

$$\leq C \cdot \sum_{\text{correct } j} \|V_j\|$$

$$\|B_k\|^r \leq C \cdot \sum_{r_1 + \cdots + r_{n-f} = r} \|V_1\|^{r_1} \cdots \|V_{n-f}\|^{r_{n-f}}.$$

Since the $V_i$'s are independent, we finally obtain that $\mathbb{E}\|\text{KR}\|^r$ is bounded above by a linear combination of terms of the form $\mathbb{E}\|V_1\|^{r_1} \cdots \mathbb{E}\|V_{n-f}\|^{r_{n-f}} = \mathbb{E}\|G\|^{r_1} \cdots \mathbb{E}\|G\|^{r_{n-f}}$ with $r_1 + \cdots + r_{n-f} = r$. This completes the proof of condition *(ii)*. □

## 3   Convergence Analysis

**Proposition 2.** *Assume that* (i) *the cost function $Q$ is three times differentiable with continuous derivatives, and is non-negative, $Q(x) \geq 0$;* (ii) *the learning rates satisfy $\sum_t \gamma_t = \infty$ and $\sum_t \gamma_t^2 < \infty$;* (iii) *the gradient estimator satisfies $\mathbb{E}G(x,\xi) = \nabla Q(x)$ and $\forall r \in \{2,\ldots,4\}$, $\mathbb{E}\|G(x,\xi)\|^r \leq A_r + B_r\|x\|^r$ for some constants $A_r, B_r$;* (iv) *there exists a constant $0 \leq \alpha < \pi/2$ such that for all $x$*

$$\eta(n,f) \cdot \sqrt{d} \cdot \sigma(x) \leq \|\nabla Q(x)\| \cdot \sin \alpha;$$

(v) *finally, beyond a certain horizon, $\|x\|^2 \geq D$, there exist $\epsilon > 0$ and $0 \leq \beta < \pi/2 - \alpha$ such that*

$$\|\nabla Q(x)\| \geq \epsilon > 0$$

$$\frac{\langle x, \nabla Q(x) \rangle}{\|x\| \cdot \|\nabla Q(x)\|} \geq \cos \beta.$$

*Then the sequence of gradients $\nabla Q(x_t)$ converges almost surely to zero.*

*Proof.* For the sake of simplicity, we write $\text{KR}_t = \text{KR}(V_1^t, \ldots, V_n^t)$. Before proving the main claim of the proposition, we first show that the sequence $x_t$ is almost surely globally confined within the region $\|x\|^2 \leq D$.

*(Global confinement).*   Let $u_t = \phi(\|x_t\|^2)$ where

$$\phi(a) = \begin{cases} 0 & \text{if } a < D \\ (a-D)^2 & \text{otherwise} \end{cases}$$

Note that

$$\phi(b) - \phi(a) \leq (b-a)\phi'(a) + (b-a)^2. \tag{1}$$

This becomes an equality when $a, b \geq D$. Applying this inequality to $u_{t+1} - u_t$ yields

$$u_{t+1} - u_t \leq \left(-2\gamma_t \langle x_t, \text{KR}_t \rangle + \gamma_t^2 \|\text{KR}_t\|^2\right) \cdot \phi'(\|x_t\|^2)$$
$$+ 4\gamma_t^2 \langle x_t, \text{KR}_t \rangle^2 - 4\gamma_t^3 \langle x_t, \text{KR}_t \rangle \|\text{KR}_t\|^2 + \gamma_t^4 \|\text{KR}_t\|^4$$
$$\leq -2\gamma_t \langle x_t, \text{KR}_t \rangle \phi'(\|x_t\|^2) + \gamma_t^2 \|\text{KR}_t\|^2 \phi'(\|x_t\|^2)$$
$$+ 4\gamma_t^2 \|x_t\|^2 \|\text{KR}_t\|^2 + 4\gamma_t^3 \|x_t\| \|\text{KR}_t\|^3 + \gamma_t^4 \|\text{KR}_t\|^4.$$

Let $\mathcal{P}_t$ denote the $\sigma$-algebra encoding all the information up to round $t$. Taking the conditional expectation with respect to $\mathcal{P}_t$ yields

$$\mathbb{E}\left(u_{t+1} - u_t | \mathcal{P}_t\right) \leq -2\gamma_t \langle x_t, \mathbb{E}\text{KR}_t \rangle + \gamma_t^2 \mathbb{E}\left(\|\text{KR}_t\|^2\right) \phi'(\|x_t\|^2)$$
$$+ 4\gamma_t^2 \|x_t\|^2 \mathbb{E}\left(\|\text{KR}_t\|^2\right) + 4\gamma_t^3 \|x_t\| \mathbb{E}\left(\|\text{KR}_t\|^3\right) + \gamma_t^4 \mathbb{E}\left(\|\text{KR}_t\|^4\right).$$

Thanks to condition *(ii)* of $(\alpha, f)$-Byzantine resilience, and the assumption on the first four moments of $G$, there exist positive constants $A_0, B_0$ such that

$$\mathbb{E}\left(u_{t+1} - u_t | \mathcal{P}_t\right) \leq -2\gamma_t \langle x_t, \mathbb{E}\text{KR}_t \rangle \phi'(\|x_t\|^2) + \gamma_t^2 \left(A_0 + B_0 \|x_t\|^4\right).$$

Thus, there exist positive constant $A, B$ such that

$$\mathbb{E}\left(u_{t+1} - u_t | \mathcal{P}_t\right) \leq -2\gamma_t \langle x_t, \mathbb{E}\mathsf{KR}_t \rangle \phi'(\|x_t\|^2) + \gamma_t^2 \left(A + B \cdot u_t\right).$$

When $\|x_t\|^2 < D$, the first term of the right hand side is null because $\phi'(\|x_t\|^2) = 0$. When $\|x_t\|^2 \geq D$, this first term is negative because (see Figure 1)

$$\langle x_t, \mathbb{E}\mathsf{KR}_t \rangle \geq \|x_t\| \cdot \|\mathbb{E}\mathsf{KR}_t\| \cdot \cos(\alpha + \beta) > 0.$$

Hence

$$\mathbb{E}\left(u_{t+1} - u_t | \mathcal{P}_t\right) \leq \gamma_t^2 \left(A + B \cdot u_t\right).$$

We define two auxiliary sequences

$$\mu_t = \prod_{i=1}^{t} \frac{1}{1 - \gamma_i^2 B} \xrightarrow[t \to \infty]{} \mu_\infty$$

$$u_t' = \mu_t u_t.$$

Note that the sequence $\mu_t$ converges because $\sum_t \gamma_t^2 < \infty$. Then

$$\mathbb{E}\left(u_{t+1}' - u_t' | \mathcal{P}_t\right) \leq \gamma_t^2 \mu_t A.$$

Consider the indicator of the positive variations of the left-hand side

$$\chi_t = \begin{cases} 1 & \text{if } \mathbb{E}\left(u_{t+1}' - u_t' | \mathcal{P}_t\right) > 0 \\ 0 & \text{otherwise} \end{cases}$$

Then

$$\mathbb{E}\left(\chi_t \cdot (u_{t+1}' - u_t')\right) \leq \mathbb{E}\left(\chi_t \cdot \mathbb{E}\left(u_{t+1}' - u_t' | \mathcal{P}_t\right)\right) \leq \gamma_t^2 \mu_t A.$$

The right-hand side of the previous inequality is the summand of a convergent series. By the quasi-martingale convergence theorem [2], this shows that the sequence $u_t'$ converges almost surely, which in turn shows that the sequence $u_t$ converges almost surely, $u_t \to u_\infty \geq 0$.

Let us assume that $u_\infty > 0$. When $t$ is large enough, this implies that $\|x_t\|^2$ and $\|x_{t+1}\|^2$ are greater than $D$. Inequality 1 becomes an equality, which implies that the following infinite sum converges almost surely

$$\sum_{t=1}^{\infty} \gamma_t \langle x_t, \mathbb{E}\mathsf{KR}_t \rangle \phi'(\|x_t\|^2) < \infty.$$

Note that the sequence $\phi'(\|x_t\|^2)$ converges to a positive value. In the region $\|x_t\|^2 > D$, we have

$$\begin{aligned} \langle x_t, \mathbb{E}\mathsf{KR}_t \rangle &\geq \sqrt{D} \cdot \|\mathbb{E}\mathsf{KR}_t\| \cdot \cos(\alpha + \beta) \\ &\geq \sqrt{D} \cdot \left(\|\nabla Q(x_t)\| - \eta(n, f) \cdot \sqrt{d} \cdot \sigma(x_t)\right) \cdot \cos(\alpha + \beta) \\ &\geq \sqrt{D} \cdot \epsilon \cdot (1 - \sin\alpha) \cdot \cos(\alpha + \beta) > 0. \end{aligned}$$

This contradicts the fact that $\sum_{t=1}^{\infty} \gamma_t = \infty$. Therefore, the sequence $u_t$ converges to zero. This convergence implies that the sequence $\|x_t\|^2$ is bounded, i.e., the vector $x_t$ is confined in a bounded region containing the origin. As a consequence, any continuous function of $x_t$ is also bounded, such as, e.g., $\|x_t\|^2$, $\mathbb{E}\|G(x_t, \xi)\|^2$ and all the derivatives of the cost function $Q(x_t)$. In the sequel, positive constants $K_1, K_2$, etc. . . are introduced whenever such a bound is used.

*(Convergence).* We proceed to show that the gradient $\nabla Q(x_t)$ converges almost surely to zero. We define

$$h_t = Q(x_t).$$

Using a first-order Taylor expansion and bounding the second derivative with $K_1$, we obtain

$$|h_{t+1} - h_t + 2\gamma_t \langle \mathsf{KR}_t, \nabla Q(x_t) \rangle| \leq \gamma_t^2 \|\mathsf{KR}_t\|^2 K_1 \text{ a.s.}$$

Therefore

$$\mathbb{E}\left(h_{t+1} - h_t | \mathcal{P}_t\right) \leq -2\gamma_t \langle \mathbb{E}\mathsf{KR}_t, \nabla Q(x_t) \rangle + \gamma_t^2 \mathbb{E}\left(\|\mathsf{KR}_t\|^2 | \mathcal{P}_t\right) K_1. \tag{2}$$

Figure 1: Condition on the angles between $x_t$, $\nabla Q(x_t)$ and $\mathbb{E}\mathrm{KR}_t$, in the region $\|x_t\|^2 > D$.

By the properties of $(\alpha, f)$-Byzantine resiliency, this implies

$$\mathbb{E}\left(h_{t+1} - h_t | \mathcal{P}_t\right) \leq \gamma_t^2 K_2 K_1,$$

which in turn implies that the positive variations of $h_t$ are also bounded

$$\mathbb{E}\left(\chi_t \cdot (h_{t+1} - h_t)\right) \leq \gamma_t^2 K_2 K_1.$$

The right-hand side is the summand of a convergent infinite sum. By the quasi-martingale convergence theorem, the sequence $h_t$ converges almost surely, $Q(x_t) \to Q_\infty$.

Taking the expectation of Inequality 2, and summing on $t = 1, \ldots, \infty$, the convergence of $Q(x_t)$ implies that

$$\sum_{t=1}^{\infty} \gamma_t \langle \mathbb{E}\mathrm{KR}_t, \nabla Q(x_t) \rangle < \infty \text{ a.s.}$$

We now define

$$\rho_t = \|\nabla Q(x_t)\|^2.$$

Using a Taylor expansion, as demonstrated for the variations of $h_t$, we obtain

$$\rho_{t+1} - \rho_t \leq -2\gamma_t \langle \mathrm{KR}_t, \left(\nabla^2 Q(x_t)\right) \cdot \nabla Q(x_t) \rangle + \gamma_t^2 \|\mathrm{KR}_t\|^2 K_3 \text{ a.s.}$$

Taking the conditional expectation, and bounding the second derivatives by $K_4$,

$$\mathbb{E}\left(\rho_{t+1} - \rho_t | \mathcal{P}_t\right) \leq 2\gamma_t \langle \mathbb{E}\mathrm{KR}_t, \nabla Q(x_t) \rangle K_4 + \gamma_t^2 K_2 K_3.$$

The positive expected variations of $\rho_t$ are bounded

$$\mathbb{E}\left(\chi_t \cdot (\rho_{t+1} - \rho_t)\right) \leq 2\gamma_t \mathbb{E}\langle \mathbb{E}\mathrm{KR}_t, \nabla Q(x_t) \rangle K_4 + \gamma_t^2 K_2 K_3.$$

The two terms on the right-hand side are the summands of convergent infinite series. By the quasi-martingale convergence theorem, this shows that $\rho_t$ converges almost surely.

We have

$$\langle \mathbb{E}\mathrm{KR}_t, \nabla Q(x_t) \rangle \geq \left(\|\nabla Q(x_t)\| - \eta(n, f) \cdot \sqrt{d} \cdot \sigma(x_t)\right) \cdot \|\nabla Q(x_t)\|$$

$$\geq \underbrace{(1 - \sin\alpha)}_{>0} \cdot \rho_t.$$

This implies that the following infinite series converge almost surely

$$\sum_{t=1}^{\infty} \gamma_t \cdot \rho_t < \infty.$$

Since $\rho_t$ converges almost surely, and the series $\sum_{t=1}^{\infty} \gamma_t = \infty$ diverges, we conclude that the sequence $\|\nabla Q(x_t)\|$ converges almost surely to zero. $\square$

## 4   Beyond Krum

In the main paper, we presented the strongest variant of Krum: the Multi-Krum aggregation rule. We refer to this aggregation rule as *mKrum* in the following. In this section we present the other aggregation rules that we tested.

- The Medoid.

  This aggregation rule is an easily computable variant of the geometric median. As discussed in the last section of the main paper, the geometric median is known to have strong statistical robustness, however there exists no algorithm yet [1] to compute its exact value [3]. Recall that the geometric median of a set of $d$-dimensional vectors $V_1, \ldots, V_n$ is defined as follows:

  $$med(V_1, \ldots, V_n) = \arg \min_{x \in \mathbb{R}^d} \sum_{i=1}^n \|V_i - x\|$$

  The geometric median does not necessarily lie among the vectors $V_1, \ldots, V_n$. A computable alternative to the median are the medoids, which are defined as follows:

  $$medoids(V_1, \ldots, V_n) = \arg \min_{x \in \{V_1, \ldots, V_n\}} \sum_{i=1}^n \|V_i - x\|.$$

  A medoid is not unique, similarly to Krum, if more than a vector minimizes the sum, we will refer to the *Medoid* as the medoid with the smallest index.

- $1 - p$ Krum.

  In this aggregation rule, the parameter server chooses the average of the proposed vectors with probability $p$, and Krum with probability $1 - p$. Moreover, we choose $p$ to depend on the learning round. In our implementation $p_t = \frac{1}{\sqrt{t}}$, where $t$ is the round number. With such a probability, and despite the presence of Byzantine workers, $1 - p$ Krum has a similar proof of convergence as Krum: the probability of choosing Krum goes to 1 when $t \mapsto \infty$. The rational is to follow averaging in the early phases, to accelerate learning in the absence of Byzantine workers, while mostly following Krum in the later phases and guarantee Byzantine resilience [4].

# 5 Experimental Details and Additional Results

We evaluate our algorithm on a distributed framework where we set some nodes to have an adversarial behavior of two kinds: *(a) The omniscient Byzantine workers:* workers have access to all the training-set (as if they breached into the other workers share of data). Those workers compute a rather precise estimator of the true gradient, and send the opposite value multiplied by an arbitrarily large factor. *(b) The Gaussian Byzantine workers:* Byzantine workers do not compute an estimator of the gradient and send a random vector, drawn from a Gaussian distribution of which we could set the variance high enough (200) to break averaging strategies.

On this distributed framework, we train two models with non-trivial (a-priori non-Convex) loss functions: a 4-layer convolutional network (ConvNet) with a final fully connected layer, and a classical multilayer perceptron (MLP) with two hidden layers, and on two tasks: spam filtering and image classification. We use cross-validation accuracy to compare the performance of different algorithms. The focus is on the Byzantine resilience of the gradient aggregation rules and not on the performance of the models per se.

**(Replacing an MLP by a ConvNet).** In addition to what have been presented in the main paper, we see from Figure 2 that, similarly to the situation on an MLP, mKrum is, despite attacks, comparable to a non-attacked averaging. In the same veine, in Figure 3, we observe that like for an MLP, the ConvNet only requires a reasonably low batch size for Krum to perform (despite 45 % Byzantine workers) as good as a non-attacked averaging.

**(Optimizing Krum).** In Figure 4 we compare the different variants in the absence of Byzantine workers, we see that Multi-Krum is comparably fast to averaging, then comes 1-p Krum, while Krum and the Medoid are slower.

Figure 2: Comparing an averaging aggregation with 0% Byzantine workers to mKrum facing 45% omniscious Byzantine workers for the ConvNet on the MNIST dataset. The cross-validation error evolution during learning is plotted for 3 sizes of the size of the mini-batch.

Figure 3: Test accuracy after 500 rounds as a function of the mini-batch size for an averaging aggregation with 0% Byzantine workers for the ConvNet on the MNIST dataset versus mKrum facing 45% of omniscious Byzantine workers.

**Learning without Byzantine workers**

Figure 4: Evolution of cross-validation accuracy with rounds for the different aggregation rules in the absence of Byzantine workers. The model is the MLP and the task is spam filtering. The mini-batch size is 3. Averaging and mKrum are the fastest, 1-p Krum is second, Krum and the Medoid are the slowest.

**Learning with 33% Byzantine workers**

Figure 5: Evolution of cross-validation accuracy with rounds for the different aggregation rules in the presence of 33% Gaussian Byzantine workers. The model is the MLP and the task is spam filtering. The mini-batch size is 3. Multi-Krum (mKrum) outperforms all the tested aggregation rules.

In the presence of Byzantine workers (Figure 5), Krum, Medoid and 1-p Krum are similarly robust. Unsurprisingly, averaging is not resilient (no improvement over time). Multi-Krum outperforms all the tested aggregation rules.

## Footnotes

[2] $\mathbb{I}(P)$ equals 1 if the predicate $P$ is true, and 0 otherwise.

[3]The computable approximate $\epsilon$-median [1] introduces a new parameter ($\epsilon$) that should be studied with respect to the risk of biasing the gradient estimator.

[4]Remember that the parameter server never knows if there are Byzantine workers or not. The latter can behave like correct workers in the beginning and fool any fraud detection measure.