[Reviews · NeurIPS 2017]

Reviewer 1



This paper proposes a new aggregation rule for gradient vectors received from different workers in distributed stochastic gradient descent to prevent Byzantine failure. It is proved in the paper that the normal aggregation rule like averaging cannot tolerant even one such bad failures. The paper defines a concept as Byzantine resilience to evaluate the tolerance of Byzantine failures and proves that the proposed aggregation function satisfies that. The experimental results shows that the proposed method works much better than without it with more and more Byzantine workers. However, the method will slow down the computation even when there is no such workers. So it seems that it’s important to trade off the robustness and efficiency. I am not quite familiar with distributed stochastic descent. My question is more related with machine learning tasks itself. In figure 4, it seems Krum also achieves worse learning error than average. So I would be more interested in how large the difference of error could be. Is it bounded? How much performance cost we need to pay for being robust to Byzantine failures using Krum?

Reviewer 2



This paper presents a formal condition for Byzantine fault-tolerant stochastic gradient aggregation, as well as an algorithm (Krum) that satisfies the condition. Given the definition, it is straightforward to see that aggregation based on linear combination (including averaging) is not Byzantine tolerant. Meanwhile the paper gives evidence that Krum is not only tolerant in theory but reasonably so in practice as well. I found the paper to be clear, well-organized, self-contained, and altogether fairly thorough. The basic motivation is clear: the literature on distributed learning has focused on statistical assumptions and well-behaved actors, but what can we do in very pessimistic conditions? To my knowledge this is the first paper that explores this view of fault-tolerance (already established in other contexts). I did not verify the analysis. A suggestion on exposition: The draft gives a bit of unusually extensive background on SGD by illustrating an unbiased stochastic gradients via cartoon figure (fig 1). This is nice but perhaps unnecessary for a stochastic optimization audience at NIPS. With that said, I see it as the authors' choice whether to keep this or try to reallocate the space to more main technical content. My remaining remarks are about the experiments: 1. The experiments are not so well-matched to the paper's main setting, and could be significantly strengthened by looking at more modern datasets and models. Current experiments are on small-scale ("single-machine") datasets for which distributed learning is not a relevant tool. The experiments in the current draft therefore serve more as a simulation or benchmark. In that sense, they do successfully show intriguing phenomena, but it is not shown that this behavior generalizes to settings where the algorithm is more likely to be used. Since the paper considers an MLP for MNIST, one natural suggestion for the next step would be a deep convolutional net for data-augmented CIFAR-10, which could take sufficiently longer to train on one machine that it would benefit from a distributed setup. 2. Why are correct workers given a mini-batch of size 3? Without explanation, this appears to be an arbitrary choice. 3. In the "cost of resilience" experiment (fig 5), reporting the error at round 500 seems to represent an early phase of training (and note indeed that the accuracies shown are not yet all that great for the task). At a batch size of 100 (on the larger end), in 500 rounds, 500*100 = 50K examples do not comprise a full pass through the training set. The experiment would be more complete with a second plot showing (or at least a remark describing) behavior closer to convergence. 4. In the "multi-Krum" experiment (fig 6), setting the parameter m to be n - f, the number of non-Byzantine machines, seems like a strongly omniscient choice (how should we know the number of adversaries f?). The experiment would be more complete by showing other values of m (for the same f). The writing suggests that setting m provides a tradeoff, so such a plot would actually serve to show the trade-off curve.

Reviewer 3



Intro: The paper introduces a novel algorithm, called Krum, that combines partially calculated gradients in a Byzantine failure tolerant way. Such an algorithm is meant to be useful in distributed training of machine learning models on very large data sets. I am a machine learner, but not a distributed computing expert. Hence, I review the manuscript from a customer perspective. --- Strengths: * The research question is very interesting. Distributed techniques start to play a more important role in machine learning as the scales of the data sets grow exponentially. * The paper is written in a sophisticated academic language. It is also nicely structured. * The proposed algorithm is supported with rigorous theoretical argumentations. --- Weaknesses: * Results in Figure 4 are not charming. When there are no Byzantine failures, the proposed algorithm lags far behind simple averaging. It starts to show its power only after one third of the workers are Byzantine. This looks like an extremely high failure rate. Apparently, averaging will remain to be extremely competitive when the failure rates are realistically low, making the motivation of the proposed approach questionable. * While the paper introduces an idea that starts to become useful when the data set is dramatically large, the reported experiments are on extremely small data sets. MNIST has 60000 instances, spambase has 4601. The real effect of the divergence between the true and the stochastic gradient starts to become visible when the data set is large enough. With today's hardware technology, a modest workstation with a few-thousand-dollar GPU can be easily trained on MILLIONS of data points without any need for such protocols as Krum. Furthermore, whether more data points than a few million are needed depends totally on applications. In many, the performance difference between 10 Million and 100 Million is ignorably small. For a machine learner to be convinced to slow down her model for the sake of safe distributedness, the authors should pinpoint applications where there is a real need for this and report results on those very applications. * The proposed algorithm is for updating global parameters from stochastic gradients calculated on minibatches. Although this approach is fundamental to many machine learning models, the central challenge of the contemporary techniques is different. Deep neural nets require distributed computing to distribute operations across "neurons" rather than minibatches. The fact that the proposed algorithm cannot be generalized to this scenario reduces the impact potential of this work significantly. Minor point: The paper would also be stronger if it cited earlier pioneer work on distributed machine learning. One example is: C.T. Chu et al., Map-Reduce for Machine Learning on Multicore, NIPS, 2007 --- Preliminary Evaluation: While I appreciate the nice theoretical work behind this paper and that distributed machine learning is an issue of key importance, the reported results shed doubt on the usefulness of the proposed approach. --- Final Evaluation: I do acknowledge that 33% Byzantine failure rate is a standard test case for general distributed computing tasks, but here our concern is training a machine learning model. The dynamics are a lot different from "somehow" processing as large data bunches as possible. The top-priority issue is accuracy, not data size. According to Figure 4, Krum severely undermines the model accuracy if there is no attack. This literally means a machine learner will accept to use Krum only when she is ABSOLUTELY sure that i) a 10-20 million data point subset will not be sufficient for satisfactory accuracy (hence distributed computing is required), and ii) at least 33% of the nodes will act Byzantine (hence Krum is required). As a machine learner, I am trying hard but not managing to find out such a case. Essentially, it is not the readership's but the authors' duty to bring those cases to attention. This is missing in both the paper and the rebuttal. I keep my initial negative vote.